# Controlling the broadband enhanced light chirality with L-shaped dielectric metamaterials

Ufuk Kilic[1] ✉, Matthew Hilfiker[1,2], Shawn Wimer[1], Alexander Ruder[1], Eva Schubert [1], Mathias Schubert[1,3] & Christos Argyropoulos [4] ✉

The inherently weak chiroptical responses of natural materials limit their usage for controlling and enhancing chiral light-matter interactions. Recently, several nanostructures with subwavelength scale dimensions were demonstrated, mainly due to the advent of nanofabrication technologies, as a potential alternative to efficiently enhance chirality. However, the intrinsic lossy nature of metals and the inherent narrowband response of dielectric planar thin films or metasurface structures pose severe limitations toward the practical realization of broadband and tailorable chiral systems. Here, we tackle these problems by designing all-dielectric silicon-based L-shaped optical metamaterials based on tilted nanopillars that exhibit broadband and enhanced chiroptical response in transmission operation. We use an emerging bottom-up fabrication approach, named glancing angle deposition, to assemble these dielectric metamaterials on a wafer scale. The reported strong chirality and optical anisotropic properties are controllable in terms of both amplitude and operating frequency by simply varying the shape and dimensions of the nanopillars. The presented nanostructures can be used in a plethora of emerging nanophotonic applications, such as chiral sensors, polarization filters, and spin-locked nanowaveguides.

The intriguing geometrical property of chirality exists at various scales from galaxies down to biomolecules. More specifically, the concept of chirality relies on whether a molecular system cannot coincide with its mirror image using simple symmetry operations (such as inversion, rotation, and translation). This intriguing material property recently attracted enormous interest due to its significant influence in the field of photonics. The optical manifestation of chirality is known as the differential absorption of left-circularly polarized (LCP) light from that of right-circularly polarized (RCP) light, also known as circular dichroism (CD). Chiral molecules exhibit weak CD response, mainly residing in the deep ultraviolet (UV) part of the spectrum, where coherent light sources are not available[1]. In addition, the large-scale mismatch between the wavelength of incident light and the size of chiral molecules causes very weak chiroptical responses in visible or infrared (IR) wavelengths with a lack of tunable operation.

Recently, artificially engineered subwavelength scale nanostructures, also known as optical metamaterials, permitted the emergence of nonracemic response to the absorption of light's circular polarization states[2,3]. This ability led to enhanced chiroptical properties that can be applied to several emerging technological applications, such as chiral sensors[4], biomedical imaging technologies[5], drug delivery systems[6], photonic integrated circuit designs[7], and quantum information technologies[8]. Nevertheless, the majority of currently demonstrated chiral metamaterials or self-assembled nanoparticles, including thin films of enantiomer molecules, exhibit relatively weak chiroptical responses with limited control on their chiral properties

[1]Department of Electrical and Computer Engineering, University of Nebraska-Lincoln, Lincoln, NE 68588, USA. [2]Onto Innovation Inc., Wilmington, MA 01887, USA. [3]Solid State Physics and NanoLund, Lund University, P.O. Box 118, 22100 Lund, Sweden. [4]Department of Electrical Engineering, The Pennsylvania State University, University Park, PA 16803, USA. ✉e-mail: ufukkilic@unl.edu; cfa5361@psu.edu

and narrowband response mainly limited to IR frequencies, i.e., not visible or ultraviolet (UV) spectra[9,10]. Other chiral nanostructures, primarily based on single scatterers, can exhibit strong chirality but are usually dispersed in aqueous media and have limited sample area which prevent their usage in on-chip photonic integrated circuit designs[11–20]. In addition, the vast majority of proposed chiral metamaterials are fabricated based on top-down approaches, such as lithography and etching techniques. However, these fabrication methods are usually complicated and costly, which leads to practical problems in terms of reproducibility. This essentially limits their usage in practical technological applications.

Previous investigations on boosting chirality mainly focused on metal-based (plasmonic) three-dimensional (3D) or planar chiral metamaterials and metasurfaces, respectively[21–29]. Generally, the fabrication of metallic nanostructures demands sophisticated fabrication processes with higher cost. More importantly, the resulting plasmonic configurations suffer from high absorption losses. To tackle these problems, dielectric metamaterials have started attracting the interest of the chiroptics research community due to their low absorption loss relatively lower fabrication cost, and high refractive index which leads to the excitation of multipole resonance modes[14,30]. Up until now, the chirality of dielectric planar metasurfaces[31–33] and 3D metamaterials[10,14], however, is rarely investigated experimentally and their response is usually weak, difficult to be controlled, and narrowband.

In this work, we tackle these problems by experimentally demonstrating and theoretically verifying broadband-enhanced chirality generated by large-scale, ultrathin, periodic all-dielectric silicon (Si) tilted nanopillar arrays forming L-shaped metamaterials. We utilize an emerging bottom-up fabrication approach called glancing angle deposition (GLAD) that has attracted remarkable interest during recent years due to its superior and relatively simple nanofabrication process with the ability to grow 3D nanostructures over a large (i.e., wafer) scale area[14,25–30]. We experimentally demonstrate the chiroptical response of the designed L-shaped metamaterials by using spectroscopic ellipsometry-based measurements both in transmission and reflection operation. This measurement method is more accurate than the widely used and simpler Stokes polarimeter analysis[30,34–37] since it eliminates possible non-chiral artifacts in the measured CD response originating from linear dichroism or other birefringence effects. The realized large area L-shaped metamaterials exhibit among the highest and broadest experimentally measured chiral response within the wide spectral range of near-IR to UV frequencies making them advantageous compared to other relevant structures[7,8,13,18–20,38–40] The presented experimental results are verified by systematic theoretical simulations that unravel the physical mechanisms of the strong chiroptical response by visualizing the chiral near-field distributions at nanoscale and excited multipole resonances. The induced strong chirality can be accurately regulated by the nanopillars length and angle leading to a unique control to the chiroptical spectral location and amplitude. The demonstrated GLAD-based nanophotonic systems can accelerate the development of innovative chiral opto-electronic devices and boost the emerging field of chiral quantum optics[8].

## Results

### Fabrication and handedness effect on chirality

The schematic representation of the proposed L-shaped metamaterial two-step fabrication process based on the GLAD technique is presented in Fig. 1. More specifically, the fabrication is performed with an electron-beam evaporator combined with the GLAD bottom-up technique. This fabrication method is capable of creating unique and complex 3D nanostructures without the use of masks or templates, making it cost-effective and fast[14], and, as a result, superior among other nanofabrication approaches. It does not require any post- or pre-chemical sample treatment unlike the usually used single or multi-step lithography techniques[29] and is also more regulated than the randomness induced from self-assembly processes[11].

Unlike previously reported metamaterials fabricated via the GLAD technique[14,25–30], in the current work we present a very simplistic design recipe for the fabrication of chiral nanostructures that is less prone to fabrication imperfections and can achieve tunable and strong chirality. The self-shadowing ability and surface diffusion dynamics during the GLAD process result in tilted columnar Si nanostructures that form the first segment of the L-shaped metamaterial that is not chiral, as will be demonstrated later. This step is followed by a second overhanging pillar segment that is grown by using a pre-deposition rotation angle (β) which consists of the main source of chirality in the resulting L-shaped metamaterials. Interestingly, the rotation angle β also determines and controls the chirality handedness of the obtained L-shaped metamaterial. More details on the GLAD fabrication process are schematically presented in Fig. 1a and b.

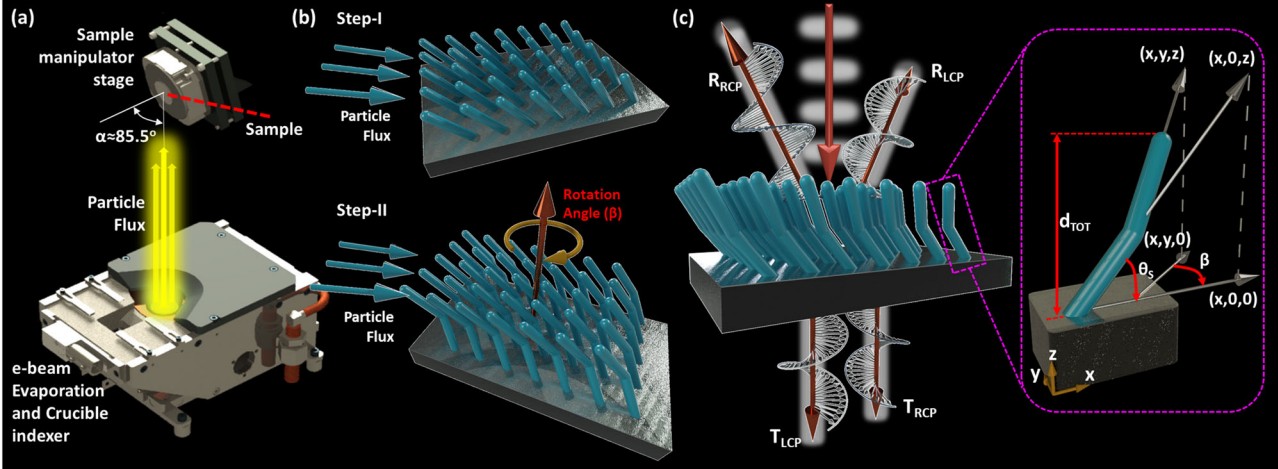

**Fig. 1 | L-shaped chiral dielectric metamaterial fabrication process. a** Electron beam assisted GLAD fabrication process. The evaporated particle flux impinges on the sample surface under oblique angles. **b** Schematic diagrams of the two-step bottom-up process to fabricate the presented chiral L-shaped metamaterials. **c** Circular polarized transmission and reflection coefficients and schematic of the proposed metamaterials. The zoomed-in illustration demonstrates the geometrical parameters of one L-shaped nanopillar, where β is the rotation angle of the second tilted nanopillar, $\theta_s$ is the initial deposition slanting angle, and $d_{TOT}$ is the total metamaterial thickness.

Kuhn's dissymmetry factor or g-factor is computed by the ratio of CD to the sum of both incident light absorption handedness. It is a frequently used unitless metric that can provide a non-biased comparison of the performance of different lossy chiral platforms. It is given by the following formula[41]:

$$g_K = 2\frac{A_- - A_+}{A_- + A_+}, \qquad (1)$$

where $A_-$ and $A_+$ are absorption of LCP and RCP light, respectively. Kuhn's dissymmetry factor calculation necessitates the measurement of both transmission and reflection responses to correctly compute the absorption in the case of transparent samples. Figure 1c represents a schematic of the L-shaped metamaterial design. It also illustrates the reflected and transmitted circular polarized waves used in Kuhn's dissymmetry factor calculations. The zoomed-in schematic in Fig. 1c points out the geometrical parameters of one L-shaped nanopillar, where $\beta$ is the rotation angle of the second tilted nanopillar, $\theta_s$ is the initial deposition slanting angle, and $d_{TOT}$ is the total metamaterial thickness. Generalized spectroscopic ellipsometry measurements by using the Mueller matrix approach[13] are utilized in the current work (more details in Supplementary Material S1) to experimentally measure the Kuhn's dissymmetry factor. This method accurately differentiates the pure chirality information from other inherent optical anisotropies in 3D samples, such as linear and circular birefringence and dichroism.

The high-resolution tilted cross section and top view scanning electron microscopy (SEM) images of the resulting right-handed (RH) and left-handed (LH) L-shaped Si metamaterials are shown in Fig. 2a and d and Fig. 2c and e, respectively. Transmission electron microscopy (TEM) analysis of the metamaterials is also performed (see Supplementary Material S2), proving that the fabricated nanostructures are mainly made of amorphous silicon. We experimentally compute by using Eq. (1) (or Eq. S2 in the Supplementary Material) the spectral evolution of Kuhn's g-factor for both RH and LH L-shaped metamaterials with the results depicted in Fig. 2b. The measured g-factor spectra in Fig. 2b demonstrate broadband and strong chiral response spanning the technologically important near-IR and visible spectra. Interestingly, depending on the rotation angle $\beta$, one can determine the handedness of the resulting nanostructures, i.e., RH metamaterials have a negative rotation angle ($\beta < 0$) while the nanostructures become LH when $\beta > 0$. The presence of structural mirror symmetry between LH and RH L-shaped metamaterials is clearly reflected in the experimentally measured positive and negative chirality results shown by red and blue lines, respectively, in Fig. 2b.

Negligible differences between the two symmetric chiral responses depicted in Fig. 2b are attributed to imperfections during the GLAD fabrication process, such as the fanning effect due to the anisotropic broadening growth of nanopillars[14,42–44]. The experimentally obtained g-factor values are very high and clearly demonstrate the strong chirality of the presented metamaterials.

## Chirality-switching response

Next, we experimentally demonstrate the complete switching of the strong chiroptical performance from non-chiral to highly chiral just by varying the rotation angle of the fabricated metamaterials. Hence, we accurately map the chirality response as a function of the metamaterial's tilted nanopillar rotation angle. These experimental results prove that the bending of the nanocolumns is the main origin of the strong chirality from the proposed metamaterial designs that are made up of two achiral nanopillar segments. Thanks to the electronic manipulation arm of the sample stage, one can precisely alter and change the rotation angle ($\beta$) prior to the second pillar subsegment deposition. We therefore fabricate a series of structures with approximately fixed thickness with an average value $d_{TOT} \approx 182$ nm but different rotation angles $\beta$: 0°, 15°, 37°, 62°, and 81°. We observe that when $\beta = 0$° there is no chirality, as demonstrated by the top $g_K$ measured spectra in Fig. 3a, despite the tilt of the entire nanorod due to the GLAD initial nanopillar segment growth. The chirality emerges and becomes stronger as $\beta$ takes non-zero values and the nanopillars start having an L-shaped geometry (middle captions in Fig. 3a). However, the chiral response starts to fade away again when $\beta$ approaches 90° (see bottom caption in Fig. 3a). This can be interpreted by considering the two tilted nanorod subsegments as optical retarder systems that operate with a 90° phase shift leading to destructive interference for the incoming circular polarized light. We also observe a small spectral shift in the $g_K$ extrema demonstrated in Fig. 3a that is attributed to the existence of small deviations in the total thickness of the fabricated samples.

In order to gain further insights into this chirality-switching phenomenon, we perform finite element method (FEM) simulations that span the entire rotation angle $\beta$ values ranging from −90° to +90°. Figure 3b shows the resulting theoretically calculated spectral evolution of $g_K$ as a function of $\beta$ angles. Note that in our theoretical modeling the metamaterial thickness is always fixed to the same value ($d_{TOT} = 180$ nm) and the $g_K$ spectra do not shift, as clearly depicted in Fig. 3b. This is different from the experimental results presented in Fig. 3a, where there is a small frequency shift in the peak chirality due to inevitable thickness variation. More details on the simulation framework are provided in the methods section. Fabrication

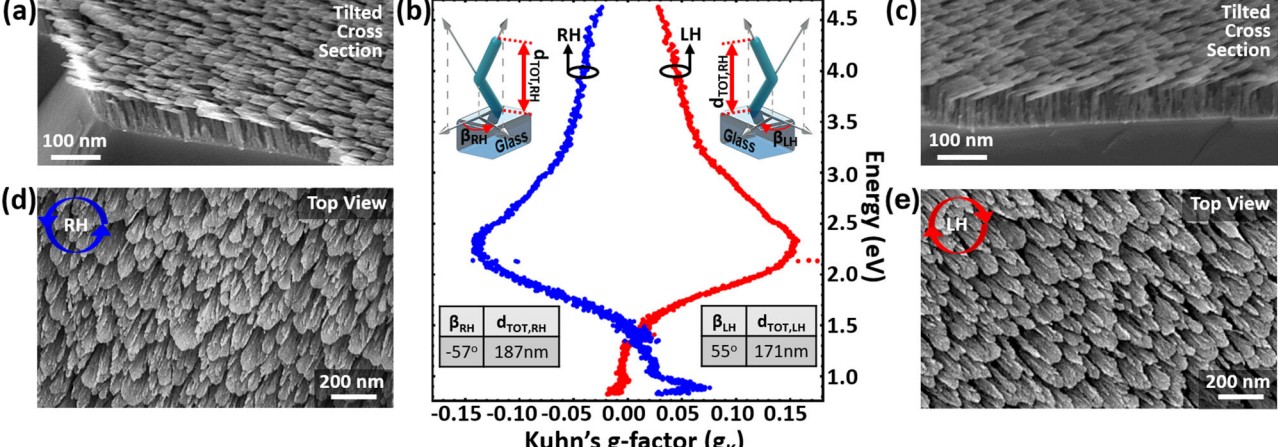

**Fig. 2 | Unraveling the strong chiroptical handedness effect.** High-resolution tilted cross section and top view SEM images of **a, d** RH and **c, e** LH L-shaped metamaterials. **b** The measured Kuhn's dissymmetry factor spectral distributions for LH (red line) and RH (blue line) metamaterials.

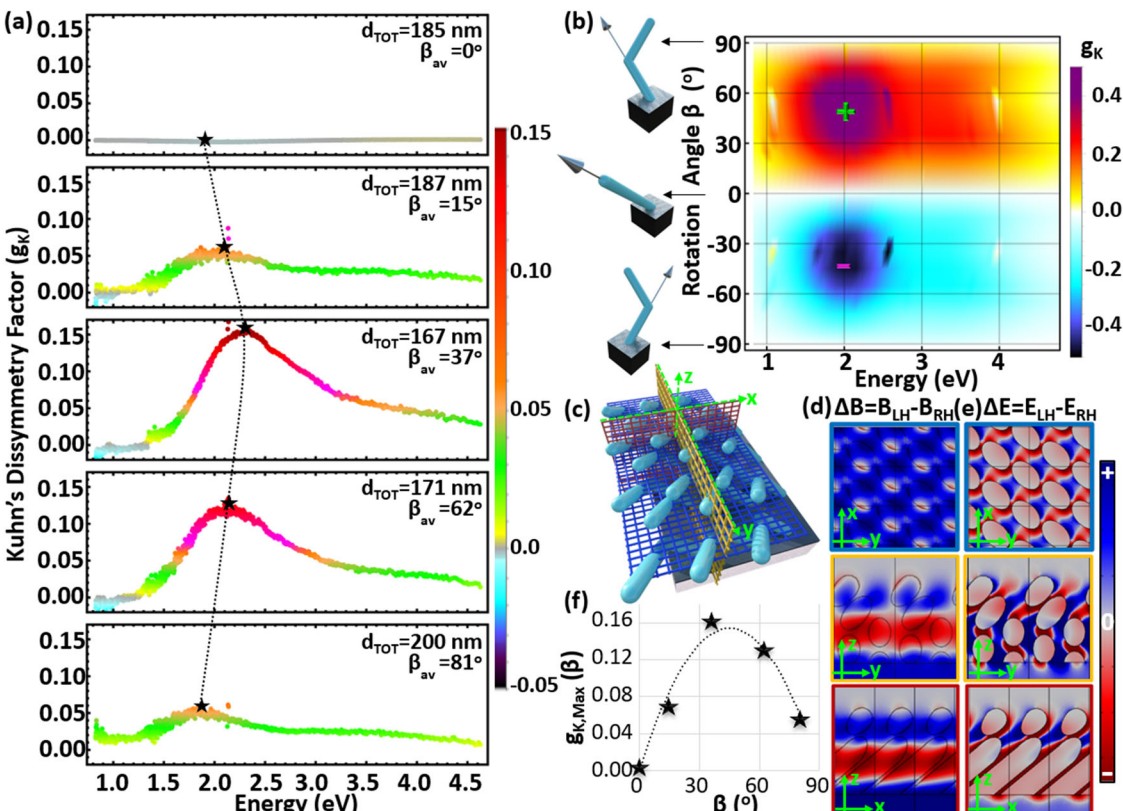

**Fig. 3 | Chirality-switching response. a** Experimentally measured and **b** theoretically computed Kuhn's dissymmetry factor spectra for different rotation angles β of the overhanging nanopillar in the L-shaped metamaterial design. **c** Schematic representation of the metamaterial array and three different planes where the asymmetric responses of the induced **d** magnetic ($\Delta \mathbf{B} = |\mathbf{B_{LH}}| - |\mathbf{B_{RH}}|$) and

(**e**) electric ($\Delta \mathbf{E} = |\mathbf{E_{LH}}| - |\mathbf{E_{RH}}|$) circular polarized fields differences are plotted. These results are computed at the photon energy indicated by the green positive sign in **b**. **f** Tracing the maximum Kuhn's dissymmetry factor values as a function of rotation angle where a dotted line is used to fit these results.

imperfections due to the GLAD method are very difficult to consider in our modeling approach. However, the presented simulation results can accurately estimate the g-factor spectral position while the amplitude of the chirality is always larger than the experimental results; this is mainly due to GLAD fabrication imperfections[14,42–44].

Full wave simulations are also able to visualize the strength of the near-field distributions induced by circularly polarized illumination which consists of important properties to better understand the induced strong chiral light-matter interactions at the nanoscale that potentially can be used for quantum optical applications[45]. These are presented by plotting the difference between the circular polarized magnetic ($\Delta | \mathbf{B_{LH}} | - | \mathbf{B_{RH}} |$) and electric ($\Delta \mathbf{E} = | \mathbf{E_{LH}} | - | \mathbf{E_{RH}} |$) field amplitudes induced by LH and RH illuminations. The results are demonstrated as two-dimensional (2D) color density slice plots demonstrated in Figs. 3(d) and 3(e), where the selected slices are schematically depicted as blue and red hatched planes in Fig. 3c. These plots were derived at the spectral location of the g-factor peak (green plus sign in Fig. 3b), i.e., strongest chiral response. The color bar displays the normalized magnitude of $\Delta \mathbf{B}$ and $\Delta \mathbf{E}$, where the blue and red colors indicate maximum positive and minimum negative differences, respectively. On the *xy* slice (indicated as a blue hatched plane in Fig. 3c), there is an opposite distribution between $\Delta \mathbf{E}$ and $\Delta \mathbf{B}$ in different nanoscale regions. While each nanopillar inner part has dark blue color for $\Delta \mathbf{B}$ (top Fig. 3d), it is red colored for $\Delta \mathbf{E}$ (top Fig. 3e). A similar behavior is observed on the *xz* slice (indicated as a red hatched plane in Fig. 3c). Interestingly, the asymmetric color distribution between $\Delta \mathbf{B}$ (bottom, Fig. 3d) and $\Delta \mathbf{E}$ (bottom, Fig. 3e) is more pronounced in the section of the overhanging nanopillar segments, and most of the first nanorod inner part segment has positive value for

both $\Delta \mathbf{B}$ (light red color) and $\Delta \mathbf{E}$ (dark red color) while their strengths are different. This response verifies that the chirality is mainly governed by the second overhanging nanopost subsegment, but the strength of chirality can be considered as the collective response of both subsegment nanopillars. The trace of maximum $g_K$ values, highlighted by star symbols in Fig. 3a, are plotted as a function of rotation angle $\beta$ in Fig. 3f. It can be concluded that the L-shaped metamaterial angle $\beta$ is the main source of the presented chirality-switching response.

## Structurally induced spectral control of chirality

With the rotation angle investigation, we observed that one can tune the chirality strength but not the frequency of chiral operation. In particular, it was derived that the g-factor extrema are not spectrally tunable as a function of rotation angle $\beta$ if the L-shaped nanorod thickness remains the same. However, the accurate spectral control of chiral resonance can have a critical role in the potential use of our proposed metamaterial designs in chiral photonic integrated circuits, nonlinear chiroptical device applications, and polarizers. With this motivation, the experimental investigations are enriched by varying other key structural parameters, such as the total thickness of the proposed metamaterial. More specifically, we fabricate the nanostructures to possess varying total thicknesses ($d_{TOT}$), but the same rotation angle is fixed to $\beta \approx 38°$ for all samples. During the GLAD process, we accurately monitor in real-time the sample's thickness evolution (see more details in the Methods section). Thereby, we fabricate metamaterials that span thickness ($d_{TOT}$) values from ~100 to ~250 nm. Figure 4a shows the experimentally measured spectral evolution of Kuhn's dissymmetry factor for each nanostructure as its

thickness is increased from bottom to top. Very strong chirality can be obtained for the largest thickness metamaterial combined with a redshift response. Based on our systematic high-resolution scanning electron microscopy (SEM) image analysis, we evaluated that the average nanopillar radius is 11 nm. Note that an increase in the nanopillar radius will cause the chiral resonance peak to redshift without substantially affecting its amplitude, as demonstrated by the theoretical results presented in Fig. S5f, while the variation in slanting angle will not change the chirality resonant frequency but will substantially vary the chirality amplitude, as measured experimentally in Fig. 3a and theoretically in Fig. 3b and S6e.

In addition, a series of theoretical simulations are performed and converted in the color density plot of Fig. 4b to demonstrate in more detail the $g_K$ spectral evolution as a function of metamaterial total thickness ($d_{TOT}$). The theoretical $g_K$ spectral response is in excellent agreement with the experimentally obtained results. The metamaterial thickness variation leads to a strong spectral control on the $g_K$ values accompanied by a narrowing in their response. The evolution of both g-factor extrema ($g_{K,Max}$) and their spectral locations ($E_{g_{K,max}}$) as a function of the total structure thickness ($d_{TOT}$) are summarized by the plot in Fig. 4c. Interestingly, when $d_{TOT}$ is above 200 nm, another higher-order resonance (≈3.35 eV) starts to emerge at larger frequency values (see Fig. 4a). A similar response is also observed in the theoretical simulations, although the new resonance is obtained at a lower part of the spectrum (≈2.65 eV). Similar to the results presented in the previous section, the simulations predict larger $g_K$ amplitudes since the theoretical modeling does not take into account surface roughness

and other fabrication imperfections induced by the GLAD process. One of our future goals will be to create GLAD structures with fewer imperfections; this will lead to even larger experimentally obtained chiroptical responses that will be closer to the amplitude of the theoretical results.

Next, we define a metric to quantitatively demonstrate the experimentally obtained and pronounced spectral tailoring of the chiroptical response and better compare our results with other relevant chiral nanostructures with variable dimensions. This metric is named spectral versatility factor, $\chi$, and is computed by the following formula: $\chi = \Delta\lambda/\Delta d = (\lambda(@d_{Max})-\lambda(@d_{Min}))/(d_{Max}-d_{Min})$, where $d_{Max}$ and $d_{Min}$ are maximum and minimum thickness values used in the experimental data shown in Fig. 4a, while $\lambda$ are the wavelengths where the chiral response spectrum (in our case characterized by $g_K$ but in other works by CD) has maximum values for the maximum and minimum thickness metamaterial designs, respectively (black star points in the upper and lower captions in Fig. 4a). Hence, the spectral versatility factor of our proposed L-shaped metamaterial platform is computed to be $\chi \approx 2.17$. To the best of our knowledge, this value is one of the highest spectral versatility factors ever measured compared with other experimental works that report tunable chiroptical activity from nanostructures in terms of either CD or $g_K$ [21–24,26,30,46,47] (see the relevant comparison in Supplementary Material Table S1). Note that a further increase in the total thickness of the GLAD fabricated metamaterials will lead to even higher $\chi$ values; this will be the subject of our future work. Based on additional systematic theoretical simulations (see Supplementary Material Fig. S6), the theoretically computed spectral

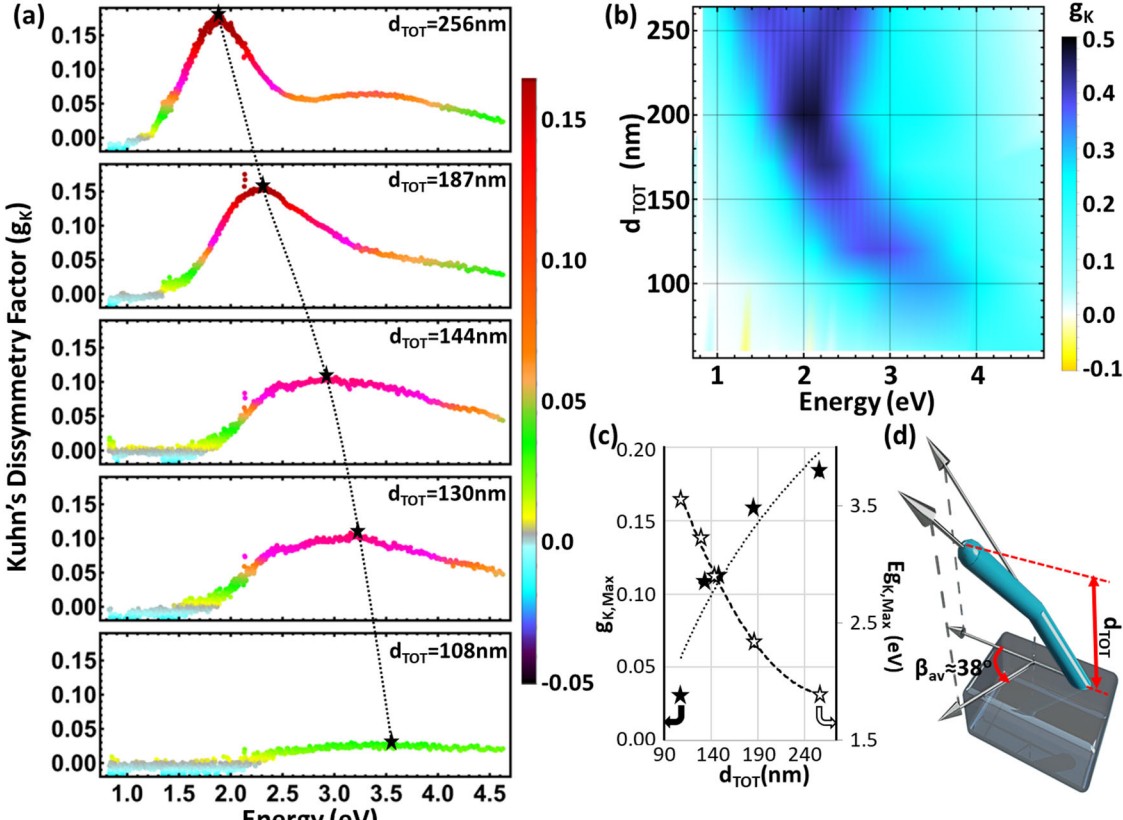

**Fig. 4 | Structurally induced spectral control of chirality. a** Experimentally measured and **b** theoretically computed Kuhn's dissymmetry factor spectra for different total thickness ($d_{TOT}$) L-shaped metamaterial designs. **c** Computed amplitude (black star) and spectral location (hollow star) of maximum Kuhn's dissymmetry values ($g_{K,Max}$). The dotted and dashed lines are used to fit these results. **d** Schematic representation of one L-shaped nanopillar. In all these results, the rotation angle of the overhanging nanopillar subsegment is kept constant and equal to 38° while the total thickness values vary. Both pillar subsegments have equal lengths.

versatility factor of the current metamaterials when the nanopillar radius increases is found to be even higher ($\chi \approx 9.84$) meaning that there is further room for improvement since larger radius nanopillars should be feasible to be realized by using the presented GLAD method.

## Metamaterial anisotropic properties analysis via Mueller matrix polarimetry

In this section, we explore the anisotropic properties of the presented metamaterial designs by using the Mueller matrix polarimetry technique. To pursue this type of investigation, we perform additional measurements based on the spectroscopic ellipsometry-based optical setup for extracting the Mueller matrix elements in transmission, as depicted in Fig. S1a. The metamaterial sample is rotated in-plane (i.e., azimuthal rotation) in this set-up by using a stage with rotation ability (blue arrows in component (8) depicted in Fig. S1a). Note that the conventional Stokes polarimetry is insufficient to reveal the anisotropic properties of the presented metamaterial and does not accurately compute the circular dichroism, as it is demonstrated in Fig. S7 in the Supplementary Material. For the characterization of complex material systems with the existence of a variety of strong optical anisotropies, similar to the studied metamaterials, the currently used Mueller matrix polarimetry is a more appropriate method to compute their accurate chiral response. This measurement method is a generalization of the conventional Stokes polarimetry technique and can compute the complete polarization behavior of an optical system, including the linear and circular dichroism and birefringence properties. More details about Mueller matrix polarimetry are provided in section S1 of the Supplementary Material.

Hence, the Mueller matrix polarimetry process is carried out at different in-plane azimuthal orientations of the sample under investigation but only in transmission mode to prove the independence of the circular dichroism from its azimuthal rotation, i.e., its isotropic chiral response. The experimentally measured spectra of the isotropic and anisotropic metamaterial properties are presented in Fig. 5, where

a wealth of information is obtained that is much more extensive than plain CD results obtained from a conventional Stokes polarimetry instrument. The metamaterials used in these studies have variable total thicknesses ($d_{TOT}$) but always fixed rotation angle $\beta = 38°$. The thickness plays a pivotal role in influencing not only the chirality response, as evident by the CD spectra illustrated in Fig. 5a, but also impacts all other optical properties. Interestingly, circular birefringence (Fig. 5b) and linear dichroism and birefringence (Fig. 5c and d, respectively) also exist in our metamaterial designs. The latter two properties (linear dichroism and birefringence) are characterized by anisotropic responses, as can be seen in the insets of Fig. 5 plots, where each metric is plotted in a fixed wavelength shown by black stars in each plot. Hence, only circular dichroism and birefringence (CD and CB responses in Figs. 5a and b, respectively) are isotropic. Note that the peak or dip of all properties redshift to lower frequencies as the metamaterial thickness is increased, similar to previously obtained g-factor results. The anisotropic nature of the proposed metamaterial designs can be hard to measure by conventional Stokes polarimetry. Moreover, the anisotropic properties compete with the isotropic circular dichroism signal leading to inaccurate CD measurements when a conventional Stokes polarimetry instrument is used, as proven in the Supplementary Material section S7.

In addition to the rotation-independent behavior of the circular dichroism and birefringence responses (refer to the inset of Figs. 5a and b, respectively), a distinct fourfold symmetry is clearly observed in the rotation-dependent evolutions of both linear birefringence and dichroism (see the inset figures in Fig. 5c and d, respectively). The inset plots in Fig. 5 correspond to the frequency depicted by the black star points in the main captions when the metamaterial thickness is maximum ($d_{TOT} = 256$ nm). A more in-depth exploration of the experimentally measured complete azimuthal rotation (ranging from 0° to 360°) dependent spectra of the metamaterial optical isotropic and anisotropic properties are included in Supplementary Material section S7 and Fig. S8. Consequently, our chiral metamaterial designs can

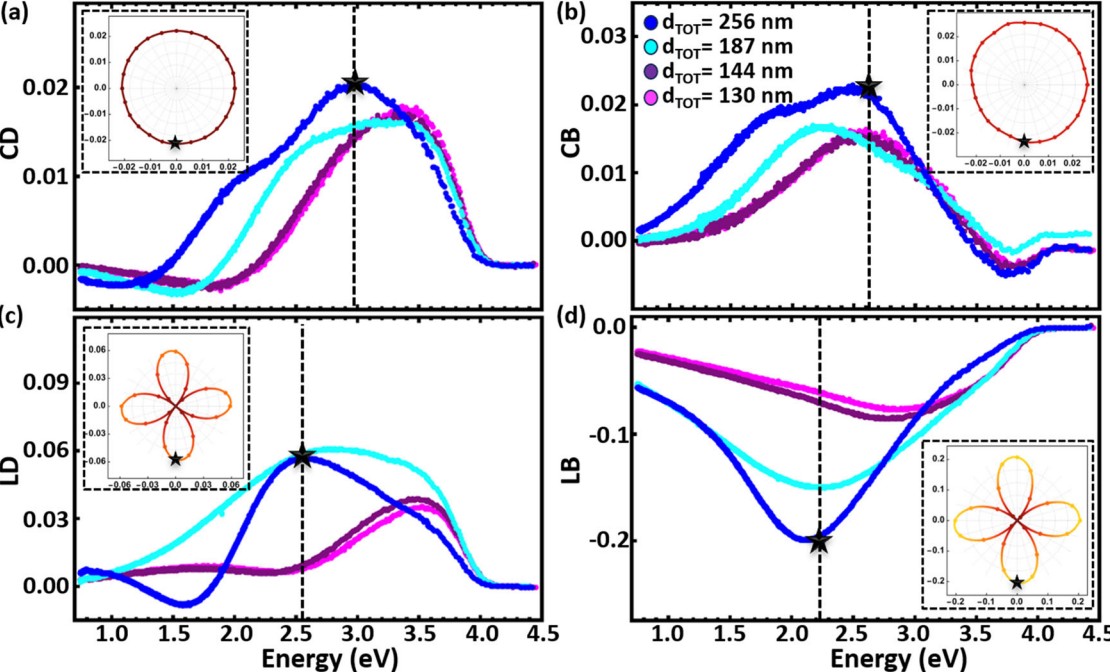

**Fig. 5 | Anisotropic properties of L-shaped metamaterials.** Mueller matrix polarimetry experimentally measured spectra of isotropic and anisotropic metamaterial properties: **a** circular dichroism **b** circular birefringence, **c** linear dichroism, and **d** linear birefringence for different total thickness ($d_{TOT}$) L-shaped metamaterial designs. These properties are measured at a fixed azimuthal orientation of the sample where the linear dichroism is maximum and linear and circular birefringence is minimum. The inset polar plots demonstrate the corresponding azimuthal orientation dependency of each optical property for a fixed frequency shown with black stars in each main plot when the metamaterial thickness is maximum ($d_{TOT} = 256$ nm). The inset plots prove that circular dichroism and birefringence are isotropic while linear dichroism and birefringence are anisotropic.

be used to applications additional to the strong circular dichroism, stemming from their anisotropic properties, such as imaging of molecules and miniaturized beam splitters[48–56].

## Theoretical analysis of chirality

The observed spectrally tailorable and broadband chiroptical response of the presented bottom-up fabricated dielectric L-shaped metamaterials is clearly induced by the overhanging pillar's rotation angle in addition to the total thickness of the nanostructure. Since the nanostructures are not 2D, i.e., have some thickness, and are made of high-index material (Si), the possibility to excite multiple electromagnetic modes exists[57]. Hence, we perform additional scattering simulations under circularly polarized illumination (LCP and RCP) to explore the effect of each resonant mode on chirality. More specifically, we decompose the total scattering coefficient ($S_{TOT, LCP}$ or $S_{TOT, RCP}$) computed for one L-shaped metamaterial unit cell to each multipole contribution, e.g., electric dipole ($S_{ED}$), magnetic dipole ($S_{MD}$), electric quadrupole ($S_{EQ}$), and magnetic quadrupole ($S_{MQ}$). More details about the used L-shaped metamaterial unit cell and scattering simulations are provided in the Methods section.

The difference between the computed total scattering coefficient under circularly polarized illumination ($S_{TOT, LCP}$ or $S_{TOT, RCP}$) is equal to the total scattering dichroism: $\Delta S_{TOT} = S_{TOT, LCP} - S_{TOT, RCP}$[57]. Since the total scattering coefficient is proportional to the sum of each multipole scattering intensity ($S_{TOT,j} \propto \sum_{i,j} S_{i,j}$[58], where $i$ is: ED, MD, EQ, MQ, and $j$ is either LCP or RCP [see Supplementary Material Eq. S6 for the expanded formula]), the dichroism for each multipole scattering component can also be defined as the difference between their LCP and RCP responses (e.g., electric dipole: $\Delta S_{ED} = S_{ED, LCP} - S_{ED, RCP}$). The summation of each scattering dichroism multipole component is equal to the total scattering dichroism: $\Delta S_{TOT} = \sum_i \Delta S_i$[57].

Figure 6a presents the spectral evolution of the computed $\Delta S_{TOT}$ together with the experimentally and theoretically obtained CD spectra computed by using absorption. We performed the electromagnetic multipole decomposition study specifically for longer tilted column length $d_{TOT} = 187$ nm ($\theta_s = 54°$, and $\beta = 38°$) to investigate the potential existence of higher order chiral resonance mode in the scattering dichroism. The obtained results follow the same trend with a single broadband resonance dip around ≈ 2.62 eV. The discrepancy between $\Delta S_{TOT}$ and CD spectra is attributed to the different unit cells used for the total scattering cross-section and absorptance calculations (further details provided in the Methods Section). By decomposing the $\Delta S_{TOT}$, we can identify the contribution of each electric/magnetic multipole with results shown in Fig. 6b, c, f, and g, where Fig. 6b and f are electric and Fig. 6c and g are magnetic dipole and quadrupole computed modes, respectively. The main contribution to the scattering dichroism comes from combining both electric and magnetic dipole responses in this long nanopillar configuration. However, we also have non-zero but much weaker chirality contributions from the electric and magnetic quadrupoles at higher energy parts of the spectra (>3 eV). While the contributions of both quadrupole modes are negligible compared to the dipole modes on the major dip in CD response, the small positive chirality in higher frequencies is mainly governed by these higher-order modes. Hence, it can be concluded that the chiral response emergence is mainly due to the electric and magnetic dipole scattering responses of the presented L-shaped metamaterials.

The far-field electric distribution profiles for both LCP and RCP total scattering are showcased in Fig. 6d computed at the photon energy value of ≈ 2.62 eV, where $\Delta S_{TOT}$ has its minimum value. The far-field total scattering patterns exhibit directional and mainly dipolar radiation profiles with the addition of various minor lobes due to the finite unit cell size used in these simulations. The obtained different valued scattering patterns under LCP and RCP excitations can be utilized to produce directional spin-polarized light emission when emitters are embedded in our metamaterial structures[59]. We also plot in Fig. 6e the corresponding normalized near-field z-component electric distribution over the surface of the L-shaped metamaterial unit cell used in the scattering simulations. The top electric field distribution in

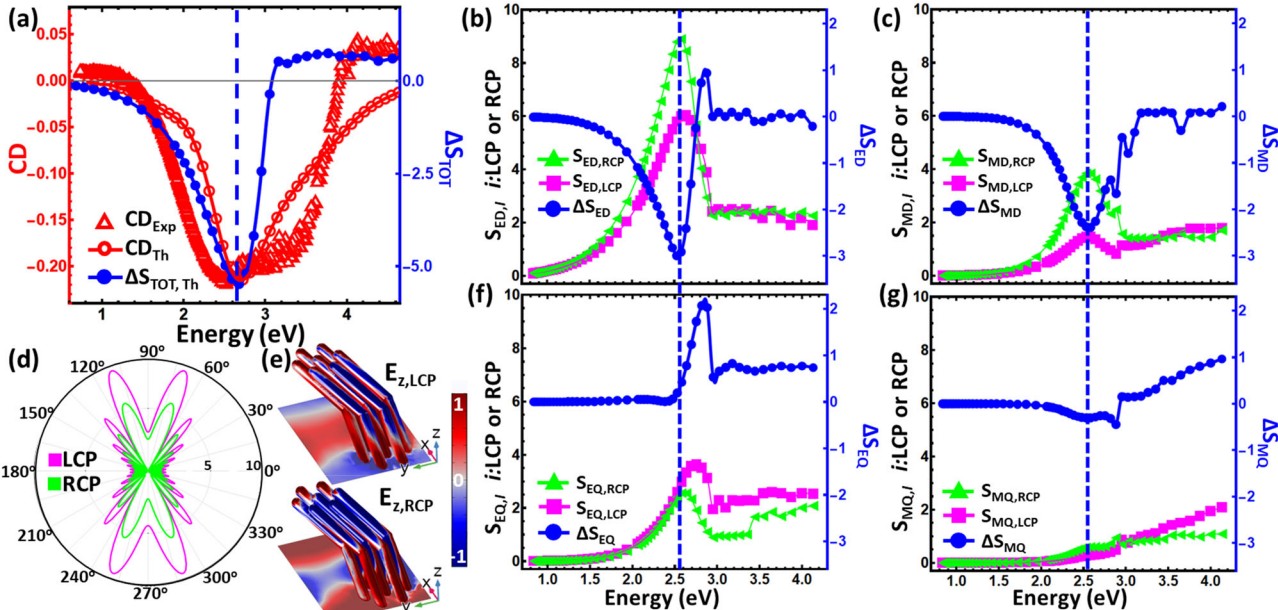

**Fig. 6 | Decomposition of total scattering dichroism into electric/magnetic multipoles. a** Experimentally measured (red triangles) circular dichroism ($CD_{Exp}$) spectrum plotted together with theoretically computed spectra of both $CD_{Th}$ (red circles) and total scattering dichroism (blue circles, $\Delta S_{TOT, Th}$). The $CD_{Exp}$ spectra is multiplied by 10 to better match the stronger $CD_{Th}$ amplitude. The vertical blue dotted lines indicate the minimum value in the $S_{TOT,Th}$ spectrum. **b** Electric and **c** magnetic dipole contributions to the L-shaped metamaterial chirality response. **f** Electric and **g** magnetic quadrupole contributions of the same L-shaped metamaterial. The vertical blue dotted lines in **b**, **c**, **f**, and **g** indicate the minimum values of $\Delta S_{ED}$, $\Delta S_{EQ}$, $\Delta S_{MD}$, and $\Delta S_{MQ}$, respectively, which always coincide at the same spectrum point. **d** 2D far field electric radiation profiles obtained from the interaction of left (magenta) and right (green) circularly polarized light with the L-shaped metamaterial unit cell. **e** 3D surface z-component electric field distribution as the incident light is left (top) or right (bottom) circular polarized.

Fig. 6e is obtained when the incident wave polarization state is LCP, while the bottom is computed for RCP illumination. These interesting results demonstrate that the chiral scattering response of the L-shaped metamaterial is not only pronounced at the far-field (Fig. 6d), but it also occurs in the nanoscale region of the near-field (Fig. 6e). Finally, further theoretical investigations on the influence of other structural parameters on the strong chiroptical response are presented in Supplementary Material Fig. S6.

## Discussion

Using the emerging bottom-up GLAD fabrication method, we develop large-scale ultrathin L-shaped dielectric metamaterials that can be compatible to on-chip photonic integrated devices. Rigorous theoretical and extensive experimental investigations are performed that clearly reveal the isotropic strong chiroptical response and optical anisotropies of this innovative class of bottom-up tunable metamaterials. The presented metamaterials achieve unprecedented control of chirality in a broadband range covering near-IR to deep UV spectra. Substantial theoretical efforts provide fundamental insights into the experimentally obtained absorption and scattering dichroism, explaining the underlying electromagnetic modes that govern the obtained strong chiral response. The presented 3D nanostructures consist of a uniquely tunable chiral metamaterial platform that can be exploited in a plethora of emerging applications, such as in the design of ultrathin polarization filters[60,61] chiral sensors[62], and directional chiral emission and lasing[63–65].

## Methods

### GLAD fabrication process

In all our fabrication efforts, an ultra-high vacuum custom-built GLAD system was utilized. During the deposition of the nanostructures, neither masks nor templates were utilized. Thanks to the high-precision sample stage manipulation arm, one can accurately control the geometry of the fabricated nanostructures during the deposition process. The incoming particle flux impinges on the sample surface under an oblique incident angle and creates randomly distributed islands on the surface that lead to a dynamic and continuous shadowing mechanism. The size and shape of the structure fabricated during the GLAD growth process are governed by the nucleation kinetics and other effects stemming from adatom surface diffusion processes. Thus, GLAD offers a fast, simple, cost-effective, large-scale area, versatile, 3D nanomorphology manufacturing process that is superior compared to other existing nanofabrication techniques.

In this study, the average base pressure of $1.0 \times 10^{-9}$ mbar was measured in the main chamber pressure gauge prior to the deposition. The glass substrate was rotated to a very large oblique angle of 85.5° compared to the source of incoming/incident vapor flux (see Fig. 1a). Having the particle flux impinge upon the sample surface at such a large oblique angle is one of the key features for achieving the presented nanostructure designs using the GLAD technique. It also has direct influence on growth morphology parameters, including initial slanting angle, radius, and density of columns. Silicon pellets with dimensions less than 3 mm and graphite liner are utilized during the electron beam-assisted GLAD fabrication process of the presented metamaterials. As schematically illustrated in Fig. 1b, this is a two-step process. Hence, following the growth of the initial pillar, the sample stage is rotated to a desired angle (i.e., rotation angle) prior to the deposition of the second overhanging nanopost.

The electron beam is controlled by the applied voltage and current. While the voltage is always fixed to 8.8 kV, the current fluctuates so that the deposition rate can be kept at a constant value, which was determined to be 0.1 Å per second. The average e-beam current value is found to be $I_{avr} \approx 180$ mA leading to a base chamber pressure of $1.2 \times 10^{-7}$ mbar. The deposition rate is controlled in situ by using a quartz crystal micro-balance sensor (QCMBS) that is located close to the sample stage. Both the QCMBS and electronic shutter system enable precise control of the metamaterial thickness during the deposition process. Finally, the Mueller matrix data is acquired in both transmission ($T$) and reflection ($R$) modes under normal incident illumination and used to compute Kuhn's dissymmetry factor (more details provided in Supplementary Material section S1.2). Conventional Stokes polarimetry was also used to measure the chirality of the metamaterial samples by using a commercially available UV-Vis CD spectropolarimeter (J-815, JASCO). The CD was measured as a function of each sample in-plane orientation with further details and results provided in the Supplementary Material section S7. The CD was found to be angle-dependent based on measurements from this instrument which does not agree with the accurate results obtained from Mueller matrix polarimetry measurements performed in our work.

### Theoretical simulations

Theoretical studies are performed by using the RF module of COMSOL Multiphysics software which accurately solves Maxwell's equations in different frequencies. We built two separate simulation setups to (i) calculate the circularly polarized absorptance and (ii) extract the scattering coefficients. The former requires the use of multiple ports that enable the S-parameters ($S_{11}$, $S_{21}$, $S_{31}$, and $S_{41}$) extraction. Using these parameters, we successfully compute the co- and cross-polarized transmittance ($T$) and reflectance ($R$) coefficients which are used in the absorptance calculations ($A = 1-R-T$). Here, $S_{21}$ and $S_{41}$ are responsible for co- (L(R)CP to L(R)CP) and cross- (L (R)CP to R(L)CP) polarized transmission coefficients, respectively. While $S_{11}$ and $S_{31}$ parameters are responsible for co- (L(R)CP to L(R)CP) and cross- (L(R)CP to R(L) CP) polarized reflection coefficients, respectively[28]. By using these coefficients, we compute the absorptance, CD, and resulted $g_K$ spectra. The simulations are always performed under normal incident illumination. They employ the amorphous state Si optical constants which are obtained from the anisotropic homogenization approach experimental extraction using reflection mode spectroscopic ellipsometry data analysis[43]. The extracted optical constants are presented in Supplementary Material S8. The presented nanorods are arranged in hexagonal formation and enclosed in a square unit cell area surrounded by periodic boundary conditions to mimic the actual GLAD fabricated design. An adaptive mesh with an element size of less than 1.5 nm is used to obtain accurate simulation results. Surface roughness and other disorders in the tilted nanopillar arrangements are neglected since they are too computationally intensive to be modeled. These simplifications are the main reason behind the quantitative difference between the theoretical and experimental results.

The second simulation setup computes the scattering coefficient ($S_{TOT}$) of each multipole contribution of the L-shaped tilted nanorods along with their scattering dichroism (see more details in Supplementary Material section S6). Unlike the previous simulation setup, the square unit cell is surrounded by perfectly matched layers (500 nm thick) to imitate an open and non-reflecting infinite domain that prevents any possible numerical artifacts on the scattering coefficient due to reflections from the unit cell boundaries. We simulate seven individual L-shaped nanorods, arranged in a hexagonal closed packing formation, that are not truncated by the boundaries. The center-to-center distance between neighboring nanopillars is taken as 36 nm and 22 nm along the x- and y-direction, respectively. Hence, the nanopillars arrangement is less dense in these scattering simulations compared to reflection/transmission modeling where the nanopillars are truncated due to their tilted geometry and thus have much smaller gaps between them.

## Data availability

The data that support the figures and other findings of this study are available from the corresponding authors upon reasonable request.

## Code availability

The codes that support the findings of this study are available from the corresponding authors upon reasonable request.

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

## Acknowledgements

This work was supported by the National Science Foundation (NSF) through EPSCoR RII Track-1: Emergent Quantum Materials and Technologies (EQUATE) Award OIA-2044049 (U.K., E.S., M.S.), NSF-CMMI 2211858 (E.S.), NSF-DMR 2224456 (E.S., C.A.), NSF-ECCS 2329940 (M.S.), Air Force Office of Scientific Research (AFOSR) FA9550-19-S-0003 (M.S.), FA9550-21-1-0259 (M.S.), and FA9550-23-1-0574 DEF (M.S.), the University of Nebraska Foundation (M.S.) and the J.A. Woollam Foundation (M.S.). All authors also thank the Systems Biology Core Facility at the University of Nebraska-Lincoln for providing access to spectropolarimeter (J-815, JASCO), the Nebraska Nanoscale Facility: National Nanotechnology Coordinated Infrastructure, and the Nebraska Center for Materials and Nanoscience.

## Author contributions

U.K., M.H., and E.S. performed GLAD deposition of the optical meta-materials. U.K. performed ALD fabrication, Mueller matrix data acquisitions, GSE analysis, and spectropolarimeter measurements. S.W., M.H., and U.K. performed HR-SEM and S/TEM image analyses. U.K, A.R., and M.S. performed the reflection mode Mueller matrix data acquisition. U.K. and C.A. performed theoretical modeling and simulations. U.K. and C.A. wrote the manuscript. E.S., M.S., and C.A. supervised the project. The manuscript was edited and approved by all authors.

## Competing interests

The authors declare no competing interests.
