## [Peer Review File · Nature Communications]

Controlling the broadband enhanced light chirality with L-shaped dielectric metamaterialsReviewers' Comments:

Reviewer #1 (Remarks to the Author):

The manuscript by Kilic and co-worker presents a technological approach to realize a dense array of chiral silicon nanowires. The approach consists of a double step glancing angle deposition and the realized structures present a Kuhn dissymmetry factor peaked in the visible. The dependence of the dissymmetry on geometrical parameters is studied theoretically and, for a set of parameters, also experimentally.

The manuscript is clearly elaborated and well written. It presents an extensive study on the optical, morphological, compositional and structural properties of the fabricated nanostructures. The problem in my opinion is the lack of novelty which makes it unsuitable for publication on Nature Communications. First of all, the state of the art lacks a comparison with significant works for the growth of 3D chiral structures, either using focused ion/electron beam induced deposition (as reviewed in Mater. Adv., 2022, 3, 186-215) which allowed demonstrating large chiroptical effects in the visible spectral range, or using on edge lithography (<https://doi.org/10.1002/adma.201203424>). Also, with respect to well known GLAD capability to realize helix nanomaterials, it is difficult to identify a significant breakthrough.

The second problem that I see is related to the claim of tunability. Indeed, the chiroptical response can be changed by changing the geometrical parameters (such as rotation angle or thickness of the metamaterial); however, tunability should be associated to a dynamically changing and possibly reversible behaviour. Therefore, I found the part of the manuscript related to this concept slightly overrated.

For these two reasons I do not find the manuscript suitable for publication in Nature Communications.

Reviewer #2 (Remarks to the Author):

This manuscript describes the fabrication of "L" shaped chiral metamaterials using the GLAD technique. There have been several studies which have used related techniques to manufacture substrates which display a chiroptical response spanning the visible to IR.

I believe the novelty in this work is that the materials display optical activity spanning the near UV to green region of the spectrum.

The question that must be addressed is whether this is sufficient to warrant publication in Nat Comm.

Unfortunately, in my opinion, the answer to this question is no. Apart from different materials this work does not have significant advance over previous work to warrant a Nat Comm. A quick scan of the literature will find several papers covering similar ground. The work is interesting and has significant merit, and I would have no issue with its publication in a journal like ACS photonics. The use of Mueller Polarimetry (MP) is very interesting and probably may have more intrinsic novelty than the materials studied, since there are relatively few studies in which the techniques is used on metamaterials.

Given this I would suggest to the authors that they make more of the MP data. For instance, it would have been interesting to discuss the linear dichroism / birefringence data ($LD(B)$ & $LD(B)'$). Given the anisotropic nature of the substrate one might expect strong birefringent effects. A comparison of the relative size of CD/LD' (± 45 degrees) would be informative. Does birefringence dominate CD , which would imply that conventional stoke polarimetry would not provide a true reflection of the chiroptical response

To summarise the work is interesting, but in my opinion does not have sufficient novelty as presented to mark it out from the crowd of similar work (apart from the MP angle which is underplayed)

Reviewer #3 (Remarks to the Author):

The authors experimentally demonstrated broadband chirality generated by large-scale, ultrathin, periodic all-dielectric silicon tilted nanopillar arrays forming L-shaped metamaterials. The fabrication is performed with an electron-beam evaporator combined with the GLAD bottom-up technique. Result shows that the strong chirality is tunable in terms of both amplitude and operating frequency by varying the shape and dimensions of the nanopillars. The experimental results and theoretical analysis are sound and the manuscript is well organized. However, the novelty of this work still remains unclear. Detailed comments are as follows.

1) Chiral metamaterial/metasurface consisting of all-dielectric or metal materials have already been reported to produce strong and tunable circular dichroism, for example, J. Li, et al., Applied Physical Letters, 118, 221110 (2021), the underlying physics was discussed intensively. Please try to be more specific on the novelties of this work.

2) Glancing angle deposition (GLAD) technique was first developed by M. Brett and K. Robbie experimentally in 1998, and it has been widely used to fabricate nanorods, nanosprings, nanocolumns, and so on. In this work, GLAD was used to fabricate the L-shaped Si nanopillar arrays. The effect of the rotation angle of the second tilted nanopillar β and the total metamaterial thickness d_{TOT} on the chiral response are investigated. What is the size of the nanopillar radius? What is the contribution of the radius and the initial deposition slanting angle θ_s to the chiral response?

3) For "tunable" claimed in this work, it is obvious to see that the Kuhn's dissymmetry factor spectra will change with the variation of the shape (the rotation angle β) and dimensions of the nanopillars (the thickness d_{TOT}). However, it does not mean "tunable".

4) On page 10, the metric named tunability factor $\chi = (d_{max} - d_{min}) / (\lambda(@d_{max}) - \lambda(@d_{min}))$ is introduced to quantitatively demonstrate the spectral tunability, where d_{max} and d_{min} are maximum and minimum thickness values used in the experimental data shown in Fig. 4(a), while λ are the wavelengths where the chiral response g_k spectrum has maximum and minimum values in the same figure. As I understand it, d_{max} is 200 nm and d_{min} is 185 nm in Fig. 4(a). However, I can only find the wavelengths where the chiral response g_k has maximum values. Please explain how to determine the value of $\lambda(@d_{min})$?

5) On page 15, the amorphous state Si optical constants should be provided for the purpose of regeneration by potential readers.

6) In Supplementary Material S2, the explanation of Figure S4(d) is very poor. I guess this figure is used to assist in testing HAADF images of structures made of different materials. Please add the clarification.

7) In Supplementary Material, the number of Eq. S(5) is missing. In addition, please unify the form of equation numbering as (S1), (S2), (S3) ... or S1, S2, S3....

Response to the reviewers' comments on the manuscript (NCOMMS-23-22678) entitled "Controlling the broadband enhanced light chirality with L-shaped dielectric metamaterials" submitted to Nature Communications

We would like to thank the three reviewers for carefully reading our paper and their constructive comments. We have revised our paper according to all reviewers' comments. We would like to stress that all the comments and suggestions were to the point and have certainly improved the quality of our manuscript and further boosted the strength and impact of the presented study. In the following, we address in detail those comments raised by the reviewers and we outline the changes in our manuscript aimed at clarifying all these relevant points.

Reviewers' comments and responses

Reviewer 1:

The manuscript by Kilic and co-worker presents a technological approach to realize a dense array of chiral silicon nanowires. The approach consists of a double step glancing angle deposition and the realized structures present a Kuhn dissymmetry factor peaked in the visible. The dependence of the dissymmetry on geometrical parameters is studied theoreticallay and, for a set of parameters, also experimentally.

The manuscript is clearly elaborated and well written. It presents an extensive study on the optical, morphological, compositional and structural properties of the fabricated nanostructures.

The problem in my opinion is the lack of novelty which makes it unsuitable for publication on Nature Communications. First of all, the state of the art lacks a comparison with significant works for the growth of 3D chiral structures, either using focused ion/electron beam induced deposition (as reviewed in Mater. Adv., 2022, 3, 186-215) which allowed demonstrating large chiroptical effects in the visible spectral range, or using on edge lithography (<https://doi.org/10.1002/adma.201203424>). Also, with respect to well known GLAD capability to realize helix nanomaterials, it is difficult to identify a significant breakthrough.

Response: We thank the reviewer for finding our paper to be clearly elaborated and well written. The reviewer argues that the weak novelty of the current work is the main concern that makes our paper unsuitable for Nature Communications. We respectfully disagree with this opinion. However, we agree that in the previous version of our work the novelty was not stressed enough, especially compared with the relevant literature based on the growth of 3D chiral nanostructures, as correctly mentioned by the reviewer. Hence, in the revised version of our work, we aim at stressing further what is new/novel in our investigation and more thoroughly compare our work with previous literature results. We also indicate why we believe our Mueller matrix polarimetry measurements are unique and will play an important role in future chiral metamaterial studies, hence, they should be presented to a broader research community that can be addressed only by the importance of a Nature Communications publication.

Firstly, we would like to stress that in the current work we propose a chiral metamaterial made of achiral building blocks. The proposed new chiral metamaterial design completely differentiates itself and is much simpler to fabricate compared to the usual helical metamaterials. Moreover, the presented new chiral metamaterial platform is fabricated by the most widely used semiconductor material (silicon), which is also another key aspect that differentiates our current work compared to the usually utilized plasmonic (metallic, silver/gold) chiral metamaterials. The use of silicon makes the current nanostructures more easily compatible to on-chip photonic integrated circuit applications, in addition to not suffering from metal induced ohmic losses. Furthermore, the proposed metamaterial designs are fabricated using a very simplistic fabrication process via a bottom-up fabrication technique (GLAD) that does not require any post- or pre-chemical sample treatment unlike the usually used single or multi-step lithography techniques, such as the edge lithography process used in the paper mentioned by the reviewer (*K. Dietrich et. al., <https://doi.org/10.1002/adma.201203424>*), where a novel-shaped plasmonic chiral nanomaterial is reported made of gold that exhibits circular dichroism in the near-infrared (not visible) spectral range. Hence, our work is totally different compared with this paper, however, we thank the reviewer for sharing it. It is now cited in our revised paper as a characteristic example of a lithography-based plasmonic chiral nanomaterial.

We also thank the reviewer for bringing to our attention the recent detailed and extensive review paper by *W. Wu and M. Pauly (Mater. Adv., 2022, 3, 186-215)* on chiral plasmonic nanostructures. Again, this review paper is focused on metallic (plasmonic) nanostructures while our currently presented metamaterials are made of semiconductors, as mentioned in the previous paragraph. Despite the different used materials, we went through all the listed metamaterials presented in this review paper, and we still believe that our design is unique compared to all these previously presented chiral nanostructures. More specifically, the uniqueness of the currently presented dielectric metamaterials stems from their design point of view as they are neither fabricated by a self-assembly of two separate/individual planar columns on top of each other nor they consist arc-like curves or made up of multiple/mixed material segments. Our proposed nanostructures are axially and spatially coherently defined, relatively well-ordered, three-dimensional and continuous nanostructures, made out of size-controllable-nano-columnar segments. Hence, we have the ability to accurately regulate the nanocolumns length which induces a unique control to the spectral location and amplitude of the induced strong chirality. Based on our very simplistic chiral metamaterial design, we observe one of the largest, broadest, and spectrally controllable chirality responses.

Moreover, thanks to the utilized combined reflection and transmission mode Mueller matrix polarimetry technique, we extracted the accurate Kuhn's dissymmetry factor values and, as a result, correct circular dichroism property. With this technique, we also extracted other optical anisotropies possessed by our metamaterial structures, demonstrating that the proposed designs also exhibit large linear birefringence and linear dichroism values that cannot be differentiated by the usually used conventional Stokes polarimetry measurements, as it is correctly argued by reviewer 2. This new investigation that is included in the revised paper version proved that the presented metamaterial structures can be used for additional applications irrelevant to their strong circular dichroism, such as imaging of molecules and miniaturized beam splitters [1-3].

Relevant to the previous comment, we also demonstrated why conventional Stokes polarimetry is an oversimplified chiroptical characterization technique to study anisotropic metamaterial designs. Note that this measurement method was utilized in the majority of previous relevant studies (including but not limited to the references [4-12]). As correctly mentioned by reviewer 2, although the use of Mueller matrix polarimetry is ideal for the accurate computation of circular dichroism, it is rarely used in previous relevant literature. The widely used conventional Stokes polarimetry is a relatively simplistic approach that unfortunately neglects the possible contamination of chirality from other angle dependent optical anisotropic properties which are irrelevant to circular dichroism that is angle independent. This leads to underestimated or overestimated erroneous angle-dependent chirality values. Hence, our study sets a new benchmark in the accurate investigation of chirality generated by metamaterial designs, correctly differentiating the anisotropic metamaterial properties compared to its isotropic circular dichroism.

However, we agree with the reviewer's opinion that the aforementioned novel aspects of our work were not presented in a correct way in the previous version of our manuscript. As a result, we made multiple changes in the revised paper to further stress the novelty of our work and added more results in the manuscript (see new Figure 5 and new Supplementary Material sections S7 and S8). All the changes are highlighted in red text in the revised main paper and Supplementary Material. Next, we summarize all the changes made to the revised manuscript following reviewer's 1 suggestions:

- 1) Our work was further compared to the relevant literature based on the growth of 3D chiral nanostructures. In the revised introduction section, we stressed that the uniqueness of the currently presented all-dielectric chiral metamaterials stems from their design point of view as they are neither fabricated by self-assembly nor lithography. We also highlighted that we can accurately regulate the nanopillars length and their angle leading to a unique control to the spectral location and amplitude of the induced strong chirality.
- 2) The complete and accurate optical activity performance of our proposed metamaterial designs is discussed in detail in the new section in the revised paper titled: "Metamaterial Anisotropic Properties Analysis via Mueller Matrix Polarimetry".
- 3) We added a new figure (Figure 5) with multiple captions which precisely demonstrate the spectral evolution of linear and circular dichroic and birefringence properties of the presented metamaterials as a function of their thicknesses.
- 4) We included an additional section in the revised Supplementary Material titled "S7. Azimuthal rotation dependent anisotropic metamaterial properties". This new section demonstrates the importance of using Mueller matrix polarimetry compared to conventional Stokes polarimetry that is widely used in commercially available instruments. More details about this section are presented later in the responses to reviewer's 2 comments.
- 5) We clearly demonstrated the angular dependence (anisotropy) of linear dichroism and birefringence as a function of azimuthal (i.e., in plane) rotation of our metamaterial samples in both revised paper and Supplementary Material.

6) We also added another section in the revised Supplementary Material titled “S8. Amorphous state Si optical constants” where we derive the amorphous silicon optical properties based on generalized spectroscopic ellipsometry data analysis.

Reviewer #1:

The second problem that I see is related to the claim of tunability. Indeed, the chiroptical response can be changed by changing the geometrical parameters (such as rotation angle or thickness of the metamaterial); however, tunability should be associated to a dynamically changing and possibly reversible behaviour. Therefore, I found the part of the manuscript related to this concept slightly overrated.

Response: We agree with the reviewer on this relevant point, and we removed the word “tunability” everywhere in the revised text. In our case, we can accurately control and tailor the chirality response spectra and amplitude based on the presented metamaterial structural parameters, as agreed by the reviewer. This indeed is not tunability, but it is unique control and tailoring of chirality by a new all-dielectric metamaterial platform. To fix this issue, we made necessary changes in the manuscript and replaced the word, “tunability” with more appropriate words (such as control and tailoring) or additional clarifications. All the manuscript changes are highlighted in red in the revised paper.

Reviewer #1:

For these two reasons I do not find the manuscript suitable for publication in Nature Communications.

Response: To sum up, we respectfully disagree with the first argument by the reviewer that our work is not novel enough. We stressed the novelty of our work in the revised version of the paper. We hope that our response will persuade the reviewer of our work’s novelty. We agree with the second point raised by the reviewer, and we have revised the entire paper to remove the tunability term. However, we strongly believe that the current control and tailoring of chirality based on simple geometrical parameter changes is unique to the currently presented new all-dielectric metamaterial platform and, as a result, another novel point of our work.

Reviewer #2:

This manuscript describes the fabrication of “L” shaped chiral metamaterials using the GLAD technique. There have been several studies which have used related techniques to manufacture substrates which display a chiroptical response spanning the visible to IR. I believe the novelty in this work is that the materials display optical activity spanning the near UV to green region of the spectrum.

The question that must be addressed is whether this is sufficient to warrant publication in Nat Comm.

Unfortunately, in my opinion, the answer to this question is no. Apart from different materials this work does not have significant advance over previous work to warrant a Nat Comm. A quick scan of the literature will find several papers covering similar ground. The work is interesting and has significant merit, and I would have no issue with its publication in a journal like ACS photonics.

The use of Mueller Polarimetry (MP) is very interesting and probably may have more intrinsic novelty than the materials studied, since there are relatively few studies in which the techniques is used on metamaterials.

Response: We would like to thank the reviewer for finding our work interesting and of significant merit. We respectfully disagree with the reviewer's opinion that our work does not have significant advancements over previous works in terms of performance, since our metamaterial exhibits a uniquely strong and controllable chiral response spanning an ultrabroadband spectrum including the near UV (3.5eV) to near-IR (1.2eV) regions. While the current demonstration of such ultrabroadband and controllable chiral response may be enough to warrant publication to Nature Communications in our opinion, we were intrigued by the reviewer's comment relevant to Mueller polarimetry investigation. We agree with the reviewer's opinion that this is another unique aspect of our work and have substantially expanded our revised paper and Supplementary Material to further discuss this rarely used but very accurate technique to characterize chirality.

Hence, following the interesting reviewer suggestions, we substantially expanded our studies on Mueller matrix polarimetry measurements which indeed are very rarely used to characterize chiral metamaterials in the literature, as correctly mentioned by the reviewer. Our results are explained in more detail later in our response letter. Briefly, we prove that the presented metamaterial has not only strong chirality (circular dichroism) but also substantial linear birefringence and linear dichroism. These properties are expected from a three-dimensional metamaterial structure but very rarely measured in the literature because the usually used conventional Stokes polarimetry cannot accurately measure chirality. Note that Stokes polarimetry can correctly measure the chirality of isotropic samples, such as chiral nanostructures randomly floating in solution, but is not an accurate approach to measure chirality of periodically arranged nanostructures formed on a substrate resulting to metamaterial designs like our study. In our work we accurately extracted both isotropic circular dichroism and birefringence and anisotropic linear dichroism and birefringence by Mueller matrix polarimetry, where we utilized both direct Mueller matrix assessment and Mueller matrix decomposition methods. We also compared our results with the ones obtained from conventional Stokes polarimetry-based instruments and stressed the differences in the measured responses.

Regarding the use of the conventional Stokes polarimetry, we added the following text at the end of the revised Methods section titled "GLAD fabrication process":

“Conventional Stokes polarimetry was also used to measure the chirality of the metamaterial samples by using a commercially available UV-Vis CD spectropolarimeter (J-815, JASCO). The CD was measured as a function of each sample in-plane orientation with further details and results

provided in the Supplementary Material. The CD was found to be angle dependent based on measurements from this instrument which does not agree with the accurate results obtained from Mueller matrix polarimetry measurements performed in our work.”

We also updated our Acknowledgements section by adding the following sentence:

“Authors also thank the Systems Biology Core Facility at University of Nebraska-Lincoln for providing access to spectropolarimeter (J-815, JASCO).”

Reviewer #2:

Given this I would suggest to the authors that they make more of the MP data. For instance, it would have been interesting to discuss the linear dichroism / birefringence data ($LD(B)$ & $LD(B)'$). Given the anisotropic nature of the substrate one might expect strong birefringent effects. A comparison of the relative size of CD / LD' (+- 45 degrees) would be informative. Does birefringence dominate CD, which would imply that conventional stoke polarimetry would not provide a true reflection of the chiroptical response.

Response: We sincerely thank the reviewer for the interesting suggestions. All of them have been addressed as explained in the following. The presented Mueller matrix polarimetry measurements applied to chiral metamaterials are new results that have substantially benefited the quality of our work. More specifically, our studies based on Mueller matrix polarimetry measurements were divided into two main components:

(i) The azimuthal (i.e., in-plane) rotation dependent analysis of chiroptical response obtained from either Mueller matrix polarimetry or conventional Stokes polarimetry methods. More specifically, we presented the measured circular dichroism as a function of both spectrum and azimuthal rotation angle obtained from these two methods to compare their accuracy. The discussion of these interesting results was included in both revised main text and new Supplementary Material section.

(ii) The accurate extraction of circular and linear dichroism and birefringence properties from Mueller matrix polarimetry. By performing Mueller matrix polarimetry, we accurately computed the Linear Birefringence (LB), Linear Dichroism (LD), Circular Birefringence (CB), and Circular Dichroism (CD) as a function of different metamaterial thickness values. The results were provided in a new section in the revised paper titled: “Metamaterial Anisotropic Properties Analysis via Mueller Matrix Polarimetry”. We also included a new figure (Figure 5) in this section with various relevant results.

Interestingly, we observed that the linear birefringence and dichroism properties are pronounced in our metamaterial design due to its three-dimensional shape and substrate leading to anisotropic properties. Back to the questions raised by the reviewer, the linear birefringence and dichroism values are comparable to the circular dichroism which necessitates the use of Mueller matrix polarimetry to obtain accurate chirality results instead of the usually employed conventional Stoke polarimetry. Moreover, we observed that all other optical activity properties of our metamaterial design are anisotropic, i.e., depending on the azimuthal orientation of sample, except, interestingly,

of the circular dichroism and birefringence. This derived isotropic chiral response is correct since the presented metamaterial unit cells carry intrinsic structural chirality due to their mirror and rotational symmetry breaking design. In addition, we followed the reviewer’s suggestions and demonstrated the spectral evolution of LD, LD’ (LD at 45° polarization), LB, LB’ (LB at 45° polarization), CB, and CD as a function of total metamaterial thickness. Some of these results are presented in the new main paper Fig. 5 and others are presented in the new Supplementary Material Fig. S8. We also presented the polar plots of each optical activity property at a selected wavelength as the azimuthal angle is varied (see inset plots in new Fig. 5) to clearly demonstrate which property is anisotropic or isotropic.

The following new section was added to the revised main paper to present the new Mueller matrix polarimetry results that were suggested by the reviewer:

“Metamaterial Anisotropic Properties Analysis via Mueller Matrix Polarimetry

In this section, we explore the anisotropic properties of the presented metamaterial designs by using the Mueller matrix polarimetry technique. To pursue this type of investigation, we perform additional measurement based on the spectroscopic ellipsometry-based optical setup for extracting the Mueller matrix elements in transmission, as depicted in Fig. S1(a). The metamaterial sample is rotated in-plane (i.e., azimuthal rotation) in this set-up by using a stage with rotation ability (blue arrows in component (8) depicted in Fig. S1(a)). Note that the conventional Stokes polarimetry is insufficient to reveal the anisotropic properties of the presented metamaterial and does not accurately compute the circular dichroism, as it is demonstrated in Fig. S7 in the Supplementary Material. For the characterization of complex material systems with the existence of a variety of strong optical anisotropies, similar to the studied metamaterials, the currently used Mueller matrix polarimetry is more appropriate method to compute their accurate chiral response. This measurement method is a generalization of the conventional Stokes polarimetry technique and can compute the complete polarization behavior of an optical system, including the linear and circular dichroism and birefringence properties. More details about Mueller matrix polarimetry are provided in section S1 of the Supplementary Material.

In this section, the Mueller matrix polarimetry process is carried out at different in-plane azimuthal orientations of the sample under investigation but only in transmission mode to prove the independence of the circular dichroism from its azimuthal rotation, i.e., its isotropic chiral response. The experimentally measured spectra of the isotropic and anisotropic metamaterial properties are presented in Fig. 5, where a wealth of information is obtained that is much more extensive than plain CD results obtained from a conventional Stokes polarimetry instrument. The metamaterials used in these studies have variable total thicknesses (d_{TOT}) but always fixed rotation angle $\beta = 38^\circ$. The thickness plays a pivotal role in influencing not only the chirality response, as evident by the CD spectra illustrated in Fig. 5(a), but also impacts all other optical properties. Interestingly, circular birefringence (Fig. 5(b)) and linear dichroism and birefringence (Figs. 5(c) and (d), respectively) also exist in our metamaterial designs. The latter two properties (linear dichroism and birefringence) are characterized by anisotropic responses, as can be seen in the insets of Fig. 5 plots, where each metric is plotted in a fixed wavelength shown by black stars in

each plot. Hence, only circular dichroism and birefringence (CD and CB responses in Figs. 5(a) and 5(b), respectively) are isotropic. Note that the peak or dip of all properties redshift to lower frequencies as the metamaterial thickness is increased, similar to previously obtained g-factor results. The anisotropic nature of the proposed metamaterial designs is impossible to measure by conventional Stokes polarimetry. Moreover, the anisotropic properties compete with the isotropic circular dichroism signal leading to inaccurate CD measurements when a conventional Stokes polarimetry instrument is used, as proven in the Supplementary Material section S7.

In addition to the rotation-independent behavior of the circular dichroism and birefringence responses (refer to the inset of Figs. 5(a) and 5(b), respectively), a distinct fourfold symmetry is clearly observed in the rotation-dependent evolutions of both linear birefringence and dichroism (see the inset figures in Figs. 5(c) and (d), respectively). The inset plots in Fig. 5 correspond to the frequency depicted by the black star points in the main captions when the metamaterial thickness is maximum ($d_{\text{TOT}}=256\text{nm}$). A more in-depth exploration of the experimentally measured complete azimuthal rotation (ranging from 0° to 360°) dependent spectra of the metamaterial optical isotropic and anisotropic properties are included in Supplementary Material section S7 and Fig. S8. Consequently, our chiral metamaterial designs can be used to applications additional to the strong circular dichroism, stemming from their anisotropic properties, such as imaging of molecules and miniaturized beam splitters [54]-[56].

Figure 5 Mueller matrix polarimetry experimentally measured spectra of isotropic and anisotropic metamaterial properties: (a) circular dichroism (b) circular birefringence, (c) linear dichroism, and (d) linear birefringence for different total thickness (d_{TOT}) L-shaped metamaterial designs. These properties are measured at a fixed azimuthal orientation of the sample where the linear dichroism is maximum and linear and circular birefringence is minimum. The inset polar plots demonstrate the corresponding azimuthal orientation dependency of each optical property for a fixed frequency shown with black stars in each main plot when the metamaterial thickness is maximum ($d_{TOT}=256\text{nm}$). The inset plots prove that circular dichroism and birefringence are isotropic while linear dichroism and birefringence are anisotropic.

We also added the following new section S7 to the revised Supplementary Material to demonstrate that the chirality response is not accurately measured by using conventional Stokes polarimetry instruments while the circular dichroism computed by the currently used Mueller matrix polarimetry is angle independent and, consequently, more accurate:

“S7. Azimuthal rotation dependent anisotropic metamaterial properties

Here, we present and discuss the chirality performance of our metamaterial design in terms of circular dichroism (CD) and other optical activity properties as a function of the sample’s azimuthal rotation. As it is discussed in the main text of the manuscript, the linear and circular optical activity properties of the current metamaterial design have comparable amplitudes that can potentially contaminate its chiral response when commercially available circular dichroism measurement instruments are used that are usually based on Stokes polarimetry. Moreover, unlike

our proposed Mueller matrix polarimetry chirality extraction that considers both transmission and reflection spectra, the commercial circular dichroism measurement instruments neglect the reflection contribution in the circular dichroism computation. The inability to differentiate linear anisotropic properties, including linear dichroism and birefringence, from the circular dichroism signal leads to an inaccurate and not precise chirality characterization from commercially available CD instruments. Moreover, commercial CD-spectropolarimeters are designed to measure isotropic samples, such as chiral nanostructures randomly dispersed in liquid solutions. Thereby, such instruments are unsuitable for the precise and accurate quantitative assessment of thin film or metamaterial chiroptical responses which can have anisotropic properties.

In addition to the CD spectra, circular birefringence (CB), linear dichroism (LD), and linear birefringence (LB) spectra are extracted by using the Mueller matrix polarimetry. While M_{32} and M_{23} elements are directly related to the circular birefringence, the linear birefringence of the sample can be computed by M_{34} and M_{43} elements. The linear dichroism is equal to M_{12} and M_{21} elements. More details about the Mueller matrix elements are presented in section S1.2 and Eq. S1. Here, we use the differential Mueller matrix formalism, which was previously applied to different organic or inorganic thin films, and compute the anisotropic optical activity properties by the formula [20]-[21]:

$$M = \ln \begin{pmatrix} 1 & m_{12} & m_{13} & m_{14} \\ m_{21} & m_{22} & m_{23} & m_{24} \\ m_{31} & m_{32} & m_{33} & m_{34} \\ m_{41} & m_{42} & m_{43} & m_{44} \end{pmatrix} = \begin{pmatrix} -\kappa & -LD & -LD' & CD \\ -LD & -\kappa & CB & LB' \\ -LD' & -CB & -\kappa & -LB \\ CD & -LB' & LB & -\kappa \end{pmatrix}. \quad (\text{S11})$$

The differential Mueller matrix M is obtained from taking the logarithm of each Mueller matrix element normalized to m_{11} and κ is the isotropic amplitude absorption. These properties are computed as a function of frequency (plotted in eV units) and in-plane (azimuthal) sample orientation, φ . The CD spectra can alternatively be represented using the ellipticity metric that is computed as follows [20]:

$$\theta_{\text{ell}} = \tan^{-1} \left(\frac{e^{\text{CD}} - 1}{e^{\text{CD}} + 1} \right). \quad (\text{S12})$$

To demonstrate the detrimental effect of other optical anisotropies on the chirality signal acquired by this type of commercial Stokes polarimetry-based instruments, we fabricate another right-handed L-shaped metamaterial on a glass substrate ($d_{\text{tot}}=120\text{nm}$, $\beta = 45^\circ$, and total substrate area= $0.6 \times 0.6\text{cm}^2$) so that the new sample can fit into the sample housing of the commercial UV-Vis CD spectropolarimeter (J-815, JASCO). We performed ellipticity measurements from 1.5 eV to 4.5 eV as a function of the sample's azimuthal rotation angle ranging from 0° to 360° with 15° step using both transmission-based Mueller matrix polarimetry method and commercial CD measurement instrument based on Stokes polarimetry. Here, we used only transmission in our Mueller matrix polarimetry measurements to achieve a fair comparison with the conventional CD spectropolarimeter that cannot measure the reflected signal from the metamaterial. The computed ellipticity plots are shown in Fig. S7(a) and S7(b) for Mueller matrix and Stokes polarimetry, respectively. The commercial Stokes polarimetry-based CD measurements in UV to near-IR range were performed using the JASCO J-815 instrument equipped with one photomultiplier tube

detector and photo acoustic modulator. The used typical scanning parameters are: i) scanning speed 10 nm/min, ii) data interval 1 nm, iii) data pitch 0.5 nm, iv) digital integration time 0.25 s, and v) accumulation one. It is important to note that our Mueller matrix polarimetry-based CD measurements are performed by using two different methods: a) decomposition of Mueller matrix elements and b) direct Mueller matrix data analysis [20]-[21], where both methods were found to produce similar results.

Figure S7. Measured ellipticity spectra as a function of azimuthal (i.e., in-plane) sample rotation obtained from (a) Mueller matrix polarimetry and (b) conventional Stokes polarimetry. (c) Ellipticity amplitude as a function of azimuthal rotation plotted at fixed 3.5 eV photon energy (white dashed lines in (a) and (b)).

While both Mueller matrix and Stokes polarimetry (commercially CD instrument) methods measure the same spectral location of ellipticity extrema and similar line shapes, the commercial instrument measures oscillations in the ellipticity spectra as a function of azimuthal rotation, which is erroneous result since the chirality or circular dichroism of the sample should not be anisotropic. The inaccurate chirality values oscillate with $\sim 122\%$ variation, as can be seen in Fig. S7(c) (red line). Chirality is expected to be independent from the azimuthal sample rotation, since it is an inherent symmetry breaking property of the structure, which results in its inability to be superimposed to its mirror image. This erroneous variation in the chirality strength is due to the inability of the commercial instrument to differentiate the purely chiral response from the other linear and circular optical activity anisotropies. Moreover, the average amplitude of ellipticity computed by conventional Stokes polarimetry is -1.25° while the Mueller matrix polarimetry measures a constant value of -1.4° , i.e., 0.25° deviation exists from the correct result. Hence, the commercially available Stokes polarimetry method is unsuitable to extract the correct chirality values for transparent, birefringent, and dichroic 3D metamaterial samples.

Next, we further explore the optical anisotropic properties of the metamaterial sample, as was derived, and depicted in Fig. 5 in the main paper. A schematic on how the azimuthal sample rotation is achieved is demonstrated in Fig. S8(a). The metric of LD measures the differential attenuation between orthogonal linear polarization states, arising from either anisotropic absorption or scattering. The metric of LB characterizes the property causing a speed difference in linearly polarized light propagation along different orthogonal axes resulting in a phase difference. Using

the transmission mode Mueller matrix polarimetry, the spectral evolutions of CD, CB, LD, LD at 45° polarization (LD'), LB, and LB at 45° polarization (LB') as a function of azimuthal sample rotation are shown in Fig. S8. We observe that CB does not oscillate as a function of the sample's azimuthal orientation, i.e., it is isotropic, similar to circular dichroism. However, the linear birefringence and dichroism are anisotropic, as was also shown in the main paper Fig. 5. Moreover, while the overall values of CB are comparable to CD, the other anisotropic property terms have larger and always oscillating values (see Fig. S8). This explains the azimuthal angle-dependent CD signal values obtained from the conventional CD spectrometer (see Fig. S7b).

Figure S8 (a) Schematic on how the azimuthal sample rotation is achieved. (b)-(f) Azimuthal rotation dependent spectra of (b) linear dichroism, (c) linear dichroism at 45° polarization, (d) circular birefringence (e) linear birefringence, (f) linear birefringence at 45° polarization.

Reviewer #2:

To summarise the work is interesting, but in my opinion does not have sufficient novelty as presented to mark it out from the crowd of similar work (apart from the MP angle which is underplayed)

Response: We thank the reviewer for the great suggestions that have substantially improved the quality of our work. We hope that the addition of Mueller matrix polarimetry results will persuade the reviewer that our work is sufficiently novel.

Reviewer #3:

The authors experimentally demonstrated broadband chirality generated by large-scale, ultrathin, periodic all-dielectric silicon tilted nanopillar arrays forming L-shaped metamaterials. The fabrication is performed with an electron-beam evaporator combined with the GLAD bottom-up technique. Result shows that the strong chirality is tunable in terms of both amplitude and operating frequency by varying the shape and dimensions of the nanopillars. The experimental results and theoretical analysis are sound and the manuscript is well organized. However, the novelty of this work still remains unclear.

Response: We thank the reviewer for finding our experimental results and theoretical analysis sound and our manuscript to be well organized. In the following, we further stress the specific novelties of our work, as suggested by the reviewer.

Reviewer #3:

Detailed comments are as follows.

1) Chiral metamaterial/metasurface consisting of all-dielectric or metal materials have already been reported to produce strong and tunable circular dichroism, for example, J. Li, et al., Applied Physical Letters, 118, 221110 (2021), the underlying physics was discussed intensively. Please try to be more specific on the novelties of this work.

2) Glancing angle deposition (GLAD) technique was first developed by M. Brett and K. Robbie experimentally in 1998, and it has been widely used to fabricate nanorods, nanosprings, nanocolumns, and so on. In this work, GLAD was used to fabricate the L-shaped si nanopillar arrays. The effect of the rotation angle of the second tilted nanopillar b and the total metamaterial thickness d_{TOT} on the chiral response are investigated. What is the size of the nanopillar radius? What is the contribution of the radius and the initial deposition slanting angle θ_s to the chiral response?

Response: We thank the reviewer for pointing out the seminal paper by M. Brett and K. Robbie on the GLAD technique. It should be noted that this first GLAD paper is not relevant to the study of the photonic properties of GLAD-based micro/nano structures. The paper only reports the versatile ability to fabricate different nanostructures based on the GLAD technique. All the GLAD-based structures presented in that particular paper or any other relevant paper in the literature (to the best of our knowledge) are different compared to the currently presented L-shaped metamaterials. In addition to the uniqueness of the presented new all-dielectric chiral metamaterial design, the detailed theoretical and experimental investigations of broadband and controllable large chiroptical response are another novel investigation of our study. The use of the custom-built GLAD technique to fabricate the proposed silicon-based metamaterial designs is crucial since the presented nanostructures can be grown over a large-scale area and, as a result, be used in different on-chip photonic integrated circuit device applications.

Even more interestingly, the use of Mueller matrix polarimetry in the chiroptical analysis is another unique aspect of our study, as was stressed by another reviewer. This rarely applied to metamaterial samples measurement method provides unique insights about their anisotropic and chiral properties. All this new information has now been included in our revised paper and Supplementary Material (see more details in our response to reviewer 2).

Regarding the specific question about the dimensions of our new L-shaped chiral metamaterials, the average nanopillar radius is 11 nm. This parameter was extracted via a systematic analysis (slanting angle, total thickness, average column radius) derived from the high resolution scanning electron microscopy (SEM) cross section images of our metamaterial designs. However, we thank the reviewer for pointing out this issue, as we indeed noticed that we did not explicitly mention this important dimension. Hence, we incorporated the following sentence in the main text:

“Based on our systematic high resolution scanning electron microscopy (SEM) image analysis, we evaluated that the average nanopillar radius is 11 nm.”

Please note that the contributions of both nanopillar radius and slanting angle on the chirality response were already studied and discussed in our Supplementary Material section S5 and Figure 3 in the main paper. Interestingly, an increase in nanopillar radius red shifts the chiral resonance peak without substantially affecting its amplitude, as demonstrated in Fig. S5(f). The variation in slanting angle does not change the chirality resonant frequency but it substantially varies the chirality amplitude, as measured experimentally in Fig. 3(a) and theoretically in Figs. 3(b) and S6(e). Interestingly, both theory and experiments agree that chirality takes its maximum value when the slanting angle is approximately 45° . Such control in the chiral response is unique to our new L-shaped metamaterial designs and has never been reported before, at least to our knowledge. We added a sentence in the main text to further stress these important points:

“Note that an increase in the nanopillar radius will cause the chiral resonance peak to redshift without substantially affecting its amplitude, as demonstrated by the theoretical results presented in Fig. S5(f), while the variation in slanting angle will not change the chirality resonant frequency but will substantially vary the chirality amplitude, as measured experimentally in Fig. 3(a) and theoretically in Figs. 3(b) and S6(e).”

Reviewer #3:

3) For “tunable” claimed in this work, it is obvious to see that the Kuhn’s dissymmetry factor spectra will be changed with the variation of the shape (the rotation angle b) and dimensions of the nanopillars (the thickness d_{TOT}). However, it does not mean “tunable”.

Response: We agree with the reviewer on this important point. We updated the entire manuscript and removed the use of the word “tunability” to prevent any further misunderstanding (see also our response to reviewer’s 1 identical comment).

Reviewer #3:

4) On page 10, the metric named tunability factor $\chi = (d_{max} - d_{min}) / (\lambda(@d_{max}) - \lambda(@d_{min}))$ is introduced to quantitatively demonstrate the spectral tunability, where d_{max} and d_{min} are maximum and minimum thickness values used in the experimental data shown in Fig. 4(a), while λ are the wavelengths where the chiral response g_K spectrum has maximum and minimum values in the same figure. As I understand it, d_{max} is 200 nm and d_{min} is 185 nm in Fig. 4(a). However, I can only find the wavelengths where the chiral response g_K has maximum values. Please explain how to determine the value of $\lambda(@d_{min})$?

Response: We are sorry about this misunderstanding which we believe is mainly caused by a typo in the previous version of our manuscript. The correct formula of the previously called tunability factor metric, which in the new paper version is named spectral versatility factor (χ), is $\chi = \Delta\lambda / \Delta d = (\lambda(@d_{Max}) - \lambda(@d_{Min})) / (d_{Max} - d_{Min})$ (inverse formula compared to the previous version), where d_{Max} and d_{Min} are maximum and minimum thickness values used in the experimental data shown in Fig. 4(a), while λ are the wavelengths where the chiral response spectra (in our case characterized by g_K but in other works by CD) have maximum values for the maximum and minimum thickness metamaterial designs, respectively, in the same figure. We computed a value of $\chi \approx 2.17$ derived from Fig. 4(a) because the maximum and minimum thicknesses of the experimentally realized metamaterials are: $d_{Max} = 256 \text{ nm}$ and $d_{Min} = 108 \text{ nm}$ (see upper and lower captions in Fig. 4(a)). The $\lambda(@d_{Max})$ and $\lambda(@d_{Min})$ are the wavelengths where the chiral response g_K spectrum has maximum values for the maximum and minimum thickness metamaterial designs shown as black star points in the upper and lower captions in Fig. 4(a). These values are computed to be $\lambda(@d_{Max}) = 345 \text{ nm}$ and $\lambda(@d_{Min}) = 666 \text{ nm}$. If we substitute all these results in the formula $\chi = \Delta\lambda / \Delta d = (\lambda(@d_{Max}) - \lambda(@d_{Min})) / (d_{Max} - d_{Min})$ then we get the computed value $\chi \approx 2.17$. The spectral versatility factor metric is used to demonstrate how much we can control the spectral location of the chiral resonant maximum values when we vary the thickness of the structure under investigation. This metric allows us to directly compare our results to other chiral nanostructures with variable dimensions. It is found that our current chiral metamaterial designs achieve one of the highest values in this type of metric, as demonstrated in Supplementary Material Table S1. We have corrected and revised the text that introduces this metric in the manuscript, and we believe that now the presentation has been improved. The relevant text is the following:

“Next, we define a metric to quantitatively demonstrate the experimentally obtained and pronounced spectral tailoring of the chiroptical response and better compare our results with other relevant chiral nanostructures with variable dimensions. This metric is named spectral versatility factor, χ , and is computed by the following formula: $\chi = \Delta\lambda/\Delta d = (\lambda(@d_{\text{Max}}) - \lambda(@d_{\text{Min}}))/(d_{\text{Max}} - d_{\text{Min}})$, where d_{Max} and d_{Min} are maximum and minimum thickness values used in the experimental data shown in Fig. 4(a), while λ are the wavelengths where the chiral response spectrum (in our case characterized by g_K but in other works by CD) has maximum values for the maximum and minimum thickness metamaterial designs, respectively (black star points in the upper and lower captions in Fig. 4(a)).”

Reviewer #3:

5) On page 15, the amorphous state Si optical constants should be provided for the purpose of regeneration by potential readers.

Response: We thank the reviewer for pointing out this issue. The amorphous state Si optical constants were added in the new Supplementary Material section S8, where the real and imaginary part of refractive index are plotted as a function of the frequency. We would like to stress that the properties of amorphous silicon are computed by our in-house generalized spectroscopic ellipsometry measurements but agree very well with amorphous silicon data provided in the literature. The extracted optical constants are used in the simulations of our work. One sentence was added in the revised main paper to cite that the amorphous state Si optical constants are presented in the Supplementary Material:

“The extracted optical constants are presented in Supplementary Material S8.”

The new Supplementary Material section S8 is:

“S8. Amorphous state silicon optical constants

The anisotropic homogenization approach experimental extraction of the optical properties of amorphous Si is utilized [22]. The results are obtained by using reflection mode spectroscopic ellipsometry data analysis with a schematic shown in Fig. S9(a). The derived amorphous state Si optical constants are presented in Fig. S9(b), where the real (red line) and imaginary part (blue line/extinction coefficient) of refractive index are plotted as a function of the frequency. While the properties of amorphous silicon are computed by our in-house generalized spectroscopic ellipsometry measurements, the results agree very well with amorphous silicon data provided in the literature [23]. The extracted optical constants are used in the simulations of our work.

Figure S9 (a) Reflection mode spectroscopic ellipsometry data analysis set-up. (b) Computed real and imaginary part of amorphous silicon refractive index spectra.”

Reviewer #3:

6) In Supplementary Material S2, the explanation of Figure S4(d) is very poor. I guess this figure is used to assist in testing HAADF images of structures made of different materials. Please add the clarification.

Response: We thank the reviewer for pointing out this issue. We added the following text to further clarify Fig. S4d:

“The latter detector provides the ability to identify the elemental composition of the nanopillar. This is demonstrated in Fig. S4(d) where we image an isolated L-shaped metamaterial nanopillar. While copper (Cu) and carbon (C) peaks stem from the TEM copper grid system and carbon meshes, we also observe a Chloride (Cl) peak due to dispersion of the structures in water. Since we use ZnO ultrathin films as our sacrificial layer, we also observe a Zinc (Zn) peak in addition to Oxygen (O). We believe that the emergence of the Sulfide (S) peak is potentially a contamination effect in the solution or water. However, it is clearly demonstrated in Fig. S4(d) that the nanopillar is made of silicon (Si).”

Reviewer #3:

7) In Supplementary Material, the number of Eq.S(5) is missing. In addition, please unify the form of equation numbering as (S1), (S2), (S3) ... or S1, S2, S3....

Response: Thank you for spotting these typos in our Supplementary Material. We added the number in Eq. S5 and unified the equations as directed by the reviewer.

To conclude, we would like to thank the reviewers for their constructive suggestions that have substantially improved the quality of our paper. We believe that the novelty of our work has been enhanced in the new version after incorporating all the suggestions by the reviewers.

References:

- [1] Zhang, H., Ni, Z., Stevens, C.E. et al. Cavity-enhanced linear dichroism in a van der Waals antiferromagnet. *Nat. Photon.* 16, 311–317 (2022).
- [2] Rodger, A., Dorrington, G., & Ang, D. L. Linear dichroism as a probe of molecular structure and interactions. *Analyst*, 141(24), 6490-6498, (2016).
- [3] Dubreuil, M., Tissier, F., Rivet, S., & Le Grand, Y. Linear diattenuation imaging of biological tissues with near infrared Mueller scanning microscopy. *Biomedical optics express*, 12(1), 41-54, (2021).
- [4] Iida, T., Ishikawa, A., Tanaka, T., Muranaka, A., Uchiyama, M., Hayashi, Y. and Tsuruta, K., Super-chiral vibrational spectroscopy with metasurfaces for high-sensitive identification of alanine enantiomers. *Applied Physics Letters*, 117(10), (2020).
- [5] Wang, Z., Teh, B.H., Wang, Y., Adamo, G., Teng, J. and Sun, H., Enhancing circular dichroism by super chiral hot spots from a chiral metasurface with apexes. *Applied Physics Letters*, 110(22), (2017).
- [6] Wang, Z., Wang, Y., Adamo, G., Teh, B.H., Wu, Q.Y.S., Teng, J. and Sun, H., A novel chiral metasurface with controllable circular dichroism induced by coupling localized and propagating modes. *Advanced Optical Materials*, 4(6), pp.883-888, (2016).
- [7] Hu, H., Sekar, S., Wu, W., Battie, Y., Lemaire, V., Arteaga, O., Poulikakos, L.V., Norris, D.J., Giessen, H., Decher, G. and Pauly, M., Nanoscale bouligand multilayers: Giant circular dichroism of helical assemblies of plasmonic 1D nano-objects. *ACS nano*, 15(8), pp.13653-13661, (2021).
- [8] Kim, J.Y., McGlothin, C., Cha, M., Pfaffenberger, Z., Emre, E.T., Choi, W., Kim, S., Biteen, J. and Kotov, N., Direct Printing of Helical Metal Arrays by Circularly Polarized Light, (2023).
- [9] Kim, H., Kim, R.M., Namgung, S.D., Cho, N.H., Son, J.B., Bang, K., Choi, M., Kim, S.K., Nam, K.T., Lee, J.W. and Oh, J.H., Ultrasensitive Near-Infrared Circularly Polarized Light Detection Using 3D Perovskite Embedded with Chiral Plasmonic Nanoparticles. *Advanced Science*, 9(5), p.2104598, (2022).
- [10] Ye, Z., Li, Z., Feng, J., Wu, C., Fan, Q., Chen, C., Chen, J. and Yin, Y., Dual-Responsive Fe₃O₄@ Polyaniline Chiral Superstructures for Information Encryption. *ACS nano*, 17(18), pp.18517-18524, (2023).

[11] Sarkar, S., Behunin, R.O. and Gibbs, J.G., Shape-dependent, chiro-optical response of uv-active, nanohelix metamaterials. *Nano letters*, 19(11), pp.8089-8096, (2019).

[12] Homma, T., Sawada, N., Ishida, T. and Tatsuma, T., Photofabrication of Chiral Plasmonic Nanostructure Arrays. *ChemNanoMat*, 9(7), p.e202300096, (2023).

REVIEWERS' COMMENTS

Reviewer #1 (Remarks to the Author):

The authors have significantly enhanced the scientific impact of the manuscript, in particular highlighting the novelty of the work which was not addressed in the previous version. It is also appreciated the more detailed analysis of Mueller Polarimetry which represents the main novelty in studying chiral metamaterials. Also the authors properly re-rated the concept of "tunability". In my opinion the revised manuscript is of adequate level for publication in Nature Communications.

Reviewer #2 (Remarks to the Author):

I wish to thank the authors for the efforts in revising the manuscript and the additional experimental (MMP data) data. The authors have addressed my comments and i recommend the manuscript for publication

Reviewer #3 (Remarks to the Author):

I have read the revised manuscript along with the comments of the other referee as well as the authors' response. The authors have made positive responses to all my concerns. However, in my view the work is still lack of novelty and the manuscript is not suitable for publication on Nature Communications.